# Uni-Perceiver-MoE: Learning Sparse Generalist Models with Conditional MoEs

**Jinguo Zhu**[1,3*†], **Xizhou Zhu**[2,3*], **Wenhai Wang**[3], **Xiaohua Wang**[1],
**Hongsheng Li**[4], **Xiaogang Wang**[4], **Jifeng Dai**[5,3✉]

[1]Xi'an Jiaotong University    [2]SenseTime Research    [3]Shanghai AI Laboratory
[4]The Chinese University of Hong Kong    [5]Tsinghua University

## Abstract

To build an artificial neural network like the biological intelligence system, recent works have unified numerous tasks into a generalist model, which can process various tasks with shared parameters and do not have any task-specific modules. While generalist models achieve promising results on various benchmarks, they have performance degradation on some tasks compared with task-specialized models. In this work, we find that interference among different tasks and modalities is the main factor to this phenomenon. To mitigate such interference, we introduce the Conditional Mixture-of-Experts (Conditional MoEs) to generalist models. Routing strategies under different levels of conditions are proposed to take both the training/inference cost and generalization ability into account. By incorporating the proposed Conditional MoEs, the recently proposed generalist model Uni-Perceiver can effectively mitigate the interference across tasks and modalities, and achieves state-of-the-art results on a series of downstream tasks via prompt tuning on 1% of downstream data. Moreover, the introduction of Conditional MoEs still holds the generalization ability of generalist models to conduct zero-shot inference on new tasks, *e.g.,* video-text retrieval and video caption. Code and pre-trained generalist models are publicly released at `https://github.com/fundamentalvision/Uni-Perceiver`.

## 1 Introduction

Generalist models that handle multiple modalities and numerous tasks have been long pursued by the machine learning community. However, previous researches [65, 89, 71] focus on developing specialized models with task-specific modules. When these models are applied to new tasks, the specifically-designed components need to be redesigned on demand and fine-tuned on sufficient downstream data. As a result, their model size increases with the number of diverse downstream tasks, conflicting with the goal of generalist models.

Recently, some pioneers [93, 79, 3, 84, 86, 62] have made preliminary attempts to build generalist models by modeling various tasks into a unified formulation. With the unified modeling, large-scale pre-training on various datasets enables the generalist models to process different downstream tasks using shared parameters. These generalist models not only achieve competitive performance on pre-training tasks [79, 3, 84, 86], but also can perform zero-shot inference on novel tasks without introducing additional parameters [93, 62].

However, compared to specialized models with specific parameters for each task, generalist models with shared parameters would suffer from the task-interference issue — different tasks with shared parameters may conflict with each other [88]. The same issue is also observed in multilingual NLP

---

*Equal contribution. †This work is done when Jinguo Zhu is an intern at Shanghai AI Laboratory. ✉Corresponding to Jifeng Dai <daijifeng@tsinghua.edu.cn>.

36th Conference on Neural Information Processing Systems (NeurIPS 2022).

models [4, 81, 83]. We argue that the task-interference issue is mainly caused by the inconsistent optimization in multi-task learning. As shown in Tab. 1, during the training phase of generalist models, the gradient directions of different tasks would be inconsistent or even opposite. Thus, if multiple tasks share parameters, the optimal update direction of the shared parameters will be uncertain, resulting in sub-optimal performance.

Allowing conflicting modalities and tasks to use separate parameters should effectively mitigate the interference issue in generalist models. Mixture of Experts (MoEs) [43, 23] provides a potential solution, which learns to activate sub-networks dynamically without introducing any task-specific modules. Nevertheless, vanilla MoEs [67] select the experts according to token representations, which suffers from high training/inference cost and neglects the information of different tasks and modalities. In this work, we argue that routing strategies of MoEs require special design when applied to generalist models for mitigating the task-interference issue.

To address the task-interference issue in generalist models, we propose Conditional Mixture-of-Experts (Conditional MoEs), which improve vanilla MoEs by introducing information under different levels of conditions, including token-level, context-level, modality-level, task-level, and predefined token attributes. In this case, vanilla MoEs is a token-level variant of our Conditional MoEs, which can be replaced by other-level variants to implement stronger generalist models. We carefully discussed the training/inference cost and generalization ability of different variants, and ablated their performances in mitigating the interference issue of generalist models. Notably, Conditional MoEs with predefined token attributes introduces 8-bit attribute embedding to describe the information of currently processed task and modalities, which demonstrate excellent computational and memory efficiency and good generalization ability.

To verify the effectiveness of Conditional MoEs, we incorporated it with the recently proposed generic perception model Uni-Perceiver [93] by replacing the linear projection in self-attention and FFN blocks with conditional MoE layers. Experiments demonstrate that, by mitigating task interference with our proposed Conditional MoEs, Uni-Perceiver can be pre-trained on various tasks jointly without performance degradation, while its generalization to other tasks can be maintained simultaneously. Our main contributions are as follows:

- We carefully analyze the task-interference issue in generalist models, and provide an explanation from the gradient direction perspective as well as a metric to quantify the issue.

- We propose Conditional MoEs to address the task-interference issue in generalist models. By introducing the information of currently processed task and modalities, Conditional MoEs effectively mitigate the interference issue, while keeping low computational and memory cost.

- Compared with previous SOTAs, our generalist model with 1% downstream data prompt tuning achieves competitive performance, while only <5% training data and <10% training cost are used. We hope this work can serve as a solid baseline for generalist models and motivate further research.

## 2   Related Works

**Specialized Models.**  Previous research focuses on building specialized models for specific tasks. CNNs [47, 26, 70] and ViTs [20, 53, 76, 80] are developed for image classification. Subsequent works re-design them to adapt to diverse downstream visual tasks, *e.g.,* object detection [63] and segmentation [15, 48]. In NLP, different architectures are specifically designed for neural machine translation [77], natural language understanding [19], and natural language generation [51]. As for vision-language tasks, previous works usually combined modality-specific encoders and representation fusion modules together [13, 54]. Recently, [89, 65, 71] integrate several specialized models into a single one to handle diverse tasks. Such integrated specialized models are equipped with multiple task-specific modules to adapt to as many downstream tasks as possible. However, these methods still follow the task-specific paradigm, which conflicts with the objective of generalist models.

**Vanilla Generalist Models.**  Vanilla generalist models handle different tasks and modalities with shared parameters. Uni-Perceiver [93] formulates various perception tasks as finding the maximum likelihood target for each input through the similarity of their representations. OFA [79], Flamingo [3] and SimVLM [84] attempt to unify different tasks into sequence-to-sequence generation. UniCORN [86] and Gato [62] further incorporate bounding box and reinforcement learning tasks into the unified formulation, respectively. These generalist models not only achieve competitive performance on pre-training tasks with shared parameters, but also can perform zero-shot inference

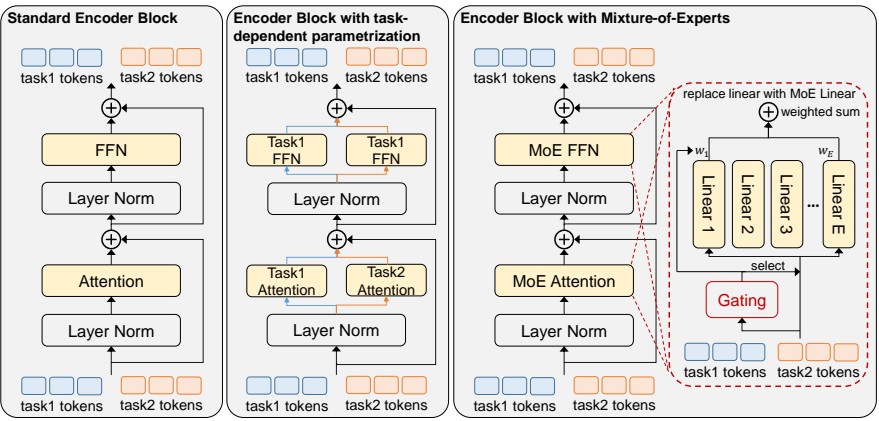

Figure 1: Comparisons of fully-shared standard encoder block, task-specific encoder block with task-dedicated parameters, and encoder block with efficient MoE parameterization.

on new tasks [62, 93]. However, these methods rarely investigate the potential interference among different modalities and tasks, which could result in the performance degradation of generalist models.

**Multi-Task Learning.** Multi-task learning [8, 17] has been widely studied in the community of vision [27, 74, 72], language [25, 16, 50] and vision-language learning [10, 55, 29]. While multi-task training enables collaboration between tasks, it may also introduce the task interference problem [81, 83, 28, 36, 72]. To mitigate the task-interference issue, some works[14, 24, 38] propose to dynamically adjust the loss weight for each task, while others [90, 49, 36] instead use task-dedicated parameters. However, methods with task-specific parameters are difficult to generalize to new tasks and do not meet the requirements of generalist models.

**Mixture of Experts (MoEs).** MoEs has shown its remarkable ability to scale neural networks [67, 43, 23, 64, 21]. [67] first proves the effectiveness of MoEs by stacking MoE layers in the LSTM models. [68, 43] further introduce this approach to Transformer architectures. [23, 40] train language models with trillion parameters successfully by utilizing simplified MoE routing strategy and efficient training techniques. There are also some works applying MoEs to CNNs for computer vision tasks [1, 85, 82]. Recently, V-MoE [64] successfully employs MoEs to ViTs, showing promising performance on many visual tasks. Task-MoE [42] focuses on applying MoEs for multilingual translation to mitigate the interference among different languages. In this work, we aim to explore MoEs under different conditions for general models.

## 3 Methodology

In this section, we first analyze the task-interference problem from the gradient direction perspective. Based on the analysis, we propose Conditional Mixture-of-Experts (Conditional MoEs) for generalist models, which introduces parameters conditioned by information of different levels to mitigate the task-interference issue with negligible overhead.

### 3.1 Task Interference

To quantify the interference of the $j$-th task on the $i$-th task, we estimate the change in loss $L_i$ of the $i$-th task, when optimizing the shared parameters $\theta$ according to the $j$-th task $L_j$ as:

$$\Delta_j L_i(x_i) \doteq \mathbb{E}_{x_j} \left( L_i(x_i; \theta) - L_i(x_i; \theta - \lambda \frac{\nabla_\theta L_j(x_j)}{\|\nabla_\theta L_j(x_j)\|}) \right) \approx \lambda \mathbb{E}_{x_j} \left( \frac{\nabla_\theta L_j(x_j)}{\|\nabla_\theta L_j(x_j)\|}^T \nabla_\theta L_i(x_i) \right), \quad (1)$$

where $x_i$ and $x_j$ are the sampled training batches of the $i$-th and $j$-th tasks, respectively, and $\lambda$ is the learning rate. Without loss of generality, we only consider the update direction ignoring the update norm. Then, the interference of the $j$-th task on the $i$-th task can be quantified as:

$$\mathcal{I}_{i,j} = \mathbb{E}_{x_i} \left( \frac{\Delta_j L_i(x_i)}{\Delta_i L_i(x_i)} \right), \quad (2)$$

Table 1: The average interference metric $\mathcal{I}_{i,j}$ of the task $j$ on the task $i$ at the 4-th/12-nd FFN blocks. To calculate the interference metric, we sample 100 batches for each tasks, and record the gradients based on the pre-trained Uni-Perceiver-Ti. The red value indicates that the task $j$ has a negative impact on the task $i$, and the green value indicates a positive impact.

(a) The 4-th FFN Block

| Task $i$ \ Task $j$ | ImgCLS (Img) | MLM (Text) | Caption (Img-Text) |
|---|---|---|---|
| ImgCLS (Img) | 1.00 | -0.57 | 1.29 |
| MLM (Text) | 0.07 | 1.00 | 0.68 |
| Caption (Img-Text) | 0.01 | 0.01 | 1.00 |

(b) The 12-nd FFN Block

| Task $i$ \ Task $j$ | ImgCLS (Img) | MLM (Text) | Caption (Img-Text) |
|---|---|---|---|
| ImgCLS (Img) | 1.00 | -2.91 | -2.45 |
| MLM (Text) | -1.65 | 1.00 | -1.05 |
| Caption (Img-Text) | -0.11 | 0.19 | 1.00 |

where the denominator is used to normalize the loss change scale. As reported in Tab. 1, we sample 100 batches for each tasks, and record the gradients to calculate the average interference metric $\mathcal{I}_{i,j}$ of the $j$-th task on the $i$-th task at the 4-th/12-nd FFN blocks. We see that, at shallow layers, the image caption task has positive impacts on image classification and masked language modeling, suggesting that cooperation between different tasks exists. While at deep layers, tasks with different optimization objectives hardly enhance each other, and the gradient directions may even opposite.

Fig. 1 summarizes three mainstream architectures for multi-task models. The first is the standard architecture [93, 31, 32] with parameters fully shared by different tasks, which suffers from task interference problem as analyzed above. The second is task-specific parameterized architecture [89, 65, 29, 71] equipped with dedicated parameters for each task. Although this architecture address the interference problem by task-specific parameters, it is difficult to generalize to new tasks that did not emerge in the training phase. Unlike the above two architectures, the Mixture-of-Experts (MoE) architecture [67, 43, 23, 40, 64] activates models sparsely according to different given inputs by selectively utilizing different subset of the model parameters. The sparse routing mechanism makes it possible to train very large generalist models, which maximizes the collaboration and meanwhile mitigates the interference problem. In this work, we focus on exploring Conditional MoEs for general models, whose experts are gated by conditions from different levels.

## 3.2 Conditional Mixture-of-Experts (Conditional MoEs)

We first describe the prototype of Conditional MoEs, and then provide its specific instantiations under different conditions, as well as the application to generalist models.

**Prototype.** Given any token $x_i$ in the input sequence $X = \{x_i\}_{i=1}^{L}$, conditional MoEs with $E$ experts firstly introduces a gate decision vector $\mathcal{G} \in \mathbb{R}^E$ that dispatches different input tokens to different experts, which is calculated as:

$$\mathcal{G} = \text{top}_k \left( \text{softmax} \left( \mathbf{W}_g \cdot R(x_i) + \epsilon \right) \right). \tag{3}$$

where $R(\cdot)$ defines a general routing strategy for gate decision, which is alternative under different conditions. $\mathbf{W}_g$ is the trainable weights in gate decision and $\epsilon$ is the noise term. The $\text{top}_k(\cdot)$ operator sets all values to be zero except the largest $k$ values. Since $\mathcal{G}$ only has $k \ll E$ non-zero values, the token $x_i$ is routed to only a small number of experts. After getting the gate decision vector $\mathcal{G}$, the corresponding output $y_i$ is the weighted combination of each expert's computation on $x_i$ as:

$$y_i = \sum_{e=1}^{E} \mathcal{G}_e \cdot \mathbf{W}_e \cdot x_i, \tag{4}$$

where $\mathbf{W}_e$ is the linear projection weights of the $e$-th expert and gate decision $\mathcal{G}_e$ determines how much the $e$-th expert contributes to the output $y_i$. Note that, experts with $\mathcal{G}_e = 0$ does not need to be computed for saving computation.

In Conditional MoEs, the routing strategy $R(\cdot)$ plays an important role in the multi-modality and multi-task training of generalist models. By sparsely activating experts according to different conditions, Conditional MoEs can mitigate the interference issue while maintaining the generality of the pretrained model. Next, we introduce variants with specific routing strategies under different conditions, as shown in Fig. 2.

**Token-Level Routing.** Similar to vanilla MoEs [67, 43, 23, 40, 64], the token-Level MoEs directly use the token representation for the routing strategy, which can be written as:

$$R_{\text{token}}(x_i) = x_i. \tag{5}$$

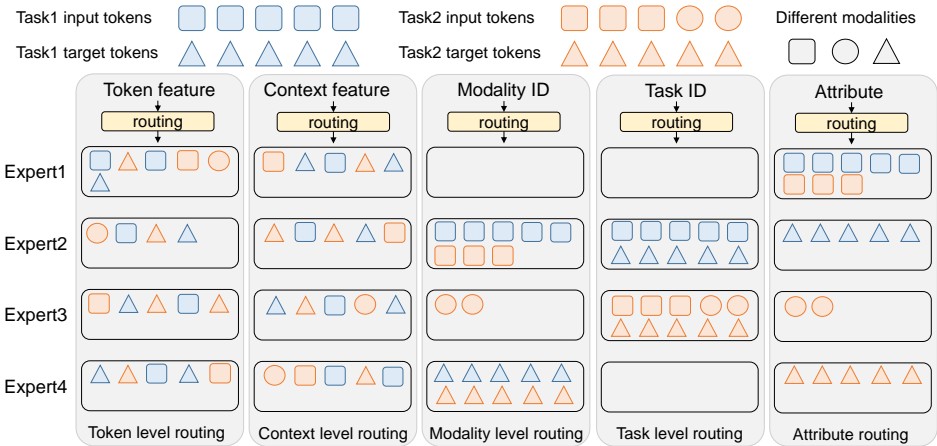

Figure 2: Comparisons of routing strategies with the top-1 gate decisions under 2-task training.

The routing strategy of token-level MoEs is an identical function, where the gate decision only depends on each token's own representation.

**Context-Level Routing.** Tokens with similar representations may appear in conflicting tasks, whose optimal expert decisions should be different to mitigate the task interference. Therefore, to help gate function making more reliable decisions, we explore the combination of global context and local token representation. The routing strategy utilizing global context can be expressed as:

$$R_{\text{context}}(x_i) = \text{concat}(x_i, \text{attnpool}(X)), \tag{6}$$

where $\text{concat}(\cdot)$ indicates the concatenation operation, $X = \{x_i\}_{i=1}^{L}$ is sequence of all tokens in the current sample, and $\text{attnpool}(\cdot)$ indicates the attention pooling operator [61].

**Modality-Level Routing.** Most current practice uses modality-specific encoders with independent parameters for different modality inputs. Inspired by this, we also explore to leverage the modality of the current token as a routing strategy:

$$R_{\text{modal}}(x_i) = \text{embed}(\text{id}_{\text{modal}}(x_i)) \tag{7}$$

Here, $\text{embed}(\cdot)$ represent the embedding layer, and $\text{id}_{\text{modal}}(\cdot)$ indicates the modality index of current token $x_i$. The routing function will assign this token to experts according to its modality embedding.

**Task-Level Routing.** In addition to modality information, task information can also be used to guide gate functions to make reliable decisions for mitigating the task interference. Similar to Eqn. (7), the routing strategy can be formulated as:

$$R_{\text{task}}(x_i) = \text{embed}(\text{id}_{\text{task}}(x_i)), \tag{8}$$

where $\text{id}_{\text{task}}(\cdot)$ is the task index of current token $x_i$. The task embedding for this task will be used to compute gate decision. Since all tokens from one task have the same task embedding, all tokens corresponding to this task will be routed to the same set of experts.

**Attribute Routing.** Among the aforementioned variants, token-level, context-level, and modality-level routing strategies only focus on input tokens but omit information about the currently processed task. While task-level routing strategy relies on task-specific ids, it limits the generalization ability to new downstream tasks. To introduce the information of currently processed task and modalities without losing the generalization ability of generalist models, we propose to introduce token attributes to assist the gate decision.

As described in Tab. 2, the attributes of the current token are represented as an 8-dimensional binary embedding, whose attributes include the modalities of current task and token (index 0~5), the causation type of the model (index 6), and the token source (index 7). As a result, the designed token attributes provide comprehensive information of currently processed task meanwhile keeping the task generalization ability. Based on the attribute embedding, the routing strategy is expressed as:

$$R_{\text{attr}}(x_i) = \text{layernorm}\left(\mathbf{W}_{\text{attr}} \cdot \text{attr}(x_i)\right). \tag{9}$$

Table 2: The 8-dimensional binary embedding used for attribute-level routing strategy. The attribute embedding is assigned to a token by checking whether the statements of the eight descriptions match the current token. For example, the attribute embedding for any token from the input sequences of image classification task should be $[1, 0, 0, 1, 1, 0, 0, 1]$. Please refer to the Appendix for detailed look-up table of attribute embeddings for all tasks in our work.

| Index | Descriptions | Yes | No |
|---|---|---|---|
| 0 | Visual modality exists in the inputs of the current task. | 1 | 0 |
| 1 | Text modality exists in the inputs of the current task. | 1 | 0 |
| 2 | Visual modality exists in the targets of the current task. | 1 | 0 |
| 3 | Text modality exists in the targets of the current task. | 1 | 0 |
| 4 | The modality of current token is visual. | 1 | 0 |
| 5 | The modality of current token is text. | 1 | 0 |
| 6 | The attention mask of the current token is causal. | 1 | 0 |
| 7 | The current token comes from the inputs, not the targets. | 1 | 0 |

Here, $\mathrm{attr}(x_i)$ is the 8-dimensional binary attribute embedding of the current token $x_i$ as described in Tab. 2. $\mathbf{W}_{\mathrm{attr}}$ is the learnable weights to transform the attribute embedding to latent representation, and $\mathrm{layernorm}(\cdot)$ denotes the layer normalization [6] for training stabilization.

**Application to Generalist Models.** Without loss of generality, we explore the application of Conditional MoEs to the generalist model Uni-Perceiver [93], which uses Transformers to handle various modalities and tasks with shared parameters. We replace linear projection layers in both self-attention and FFN blocks with Conditional-MoE layers (see Fig. 1).

### 3.3 Comparison of Conditional-MoE Variants

As illustrated in Fig. 2, among the variants of Conditional MoEs, token-level and context-level MoEs are data-dependent, while modality-level, task-level, and attribute MoEs are data-independent.

**Training and Inference Cost.** Compared to dense models with the same number of parameters, all Conditional MoE variants can significantly reduce the computational cost benefiting from the sparse routing mechanism. Due to the dependence of input data, the memory consumption of token-level and context-level MoEs is relatively high during model training, and model parallelism is required to relieve memory cost by partitioning experts across multiple devices, leading to heavy inter-device communication overhead. This problem persists when using pre-trained models for task-specific inference, where all experts need to be loaded into memory and might be activated by any token.

Different from data-dependent Conditional MoEs, data-independent variants such as modality-level, task-level, and attribute MoEs have excellent memory efficiency, since only top-$k$ experts need to be activated for all tokens with the same modality/task/attributes. Moreover, in both training and inference phase, the experts in a data-independent MoE layer can be merged into a single linear projection using reparameterization techniques. In this case, the computation cost of the network with data-independent Conditional MoEs will be equivalent to a dense model without MoEs.

**Generalization Ability.** We hope to mitigate the task-interference issue in generalist models, while keeping their generalization ability to new downstream tasks. While token-level, context-level, and modality-level MoEs without task-specific designs do not harm the generalization ability, they ignore the task-level information which is essential to resolve the task interference. Conversely, task-level routing strategy is tied to a specific task id, which is difficult to generalize to new downstream tasks. Attribute MoEs introduce predefined token attributes to comprehensively describe the information of currently processed task and modalities, which can be transferred to new downstream tasks without any task-specific modifications. This gives attribute MoEs the potential to mitigate task interference without losing generalization ability.

## 4 Experiments

In this section, we first describe our experimental setup. Then, we confirm the task-interference issue in the generalist model Uni-Perceiver [93] and ablate the the ability of different Conditional MoEs to mitigate task interference. Finally, large-scale training is conducted to verify the effectiveness of our proposed Conditional-MoEs and its generalization ability to novel tasks.

Table 3: The performance of different routing strategies for Conditional MoEs. The base model is Uni-Perceiver with BERT$_{tiny}$. We also illustrate the task-specific variant where each task has its own specialized parameters. The training and validation performance reported on three tasks: image classification on ImageNet-1K [18], image caption on COCO Caption [12], and Masked Language Modeling(MLM) on Books&Wiki. The best results within a tolerance of 1% are in **bold**.

| model | task-specific parameterization | training time | inference time | ImageNet-1k $\uparrow$acc$_{train}$ | $\uparrow$acc$_{val}$ | COCO Caption $\uparrow$acc$_{train}$ | $\uparrow$B@4$_{val}$ | MLM $\uparrow$acc$_{train}$ | $\downarrow$ppl$_{val}$ |
|---|---|---|---|---|---|---|---|---|---|
| Uni-Perceiver-Ti [93] | | 1.0× | 1.0× | 47.3 | 68.3 | 49.2 | 18.2 | 54.5 | 5.86 |
| | ✓ | 1.1× | 1.0× | **53.3** | **73.5** | **52.6** | 20.4 | **60.5** | **4.48** |
| + Conditional MoEs $_{token}$ | | 1.8× | 2.2× | **53.1** | 72.7 | **52.9** | 20.9 | 58.3 | 4.96 |
| + Conditional MoEs $_{context}$ | | 2.2× | 2.6× | 52.5 | **73.1** | **52.8** | 21.5 | 58.6 | 4.86 |
| + Conditional MoEs $_{modality}$ | | 1.4× | 1.0× | 51.7 | 72.6 | 52.1 | 21.8 | 57.5 | 5.06 |
| + Conditional MoEs $_{task}$ | | 1.4× | 1.0× | **52.9** | **73.2** | **52.7** | 21.2 | **59.9** | **4.56** |
| **+ Conditional MoEs $_{attribute}$** | | 1.4× | 1.0× | **52.8** | **73.3** | **53.1** | **23.0** | **60.0** | **4.56** |

## 4.1 Datasets

We use the same datasets in Uni-Perceiver [93] to pre-train our models[1]. Specifically, ImageNet-21k [18] is used for image classification pre-training. Kinetics-700 [37] and Moments in Time [57] are used for video classification pre-training. Language modeling task is trained on BookCorpus [94] & English Wikipedia (Books&Wiki). For language modeling with image clues and image-text retrieval, we use a combination of image-text-pair datasets: SBU Captions (SBU) [58], Visual Genome [41], COCO Caption [12], CC3M [66], CC12M [9] and YFCC [35]. Following Uni-Perceiver, Imagenet1K [18], Kinetics-400 [37], COCO Caption [12], and Flickr30k [59] are utilized to evaluate the performance of generalist models on downstream tasks. We also use two datasets that evaluate the generalization ability to novel tasks: MSVD [11] and GLUE [78]. Additionally, all dataset licenses are included in Appendix.

## 4.2 Implementation Details

We incorporate the vanilla generalist model Uni-Perceiver with Conditional MoEs for experiments with three different variants: Uni-Perceiver-Ti (Tiny), Uni-Perceiver-B (Base), and Uni-Perceiver-L (Large). Please refer to Appendix for architecture hyperparameters. If not specified, the input image resolution is set to 224×224. In each training iteration, each GPU independently samples a single task and dataset. The gradients of different GPUs are synchronized after the gradient back-propagation. We use the AdamW optimizer with a base learning rate of 0.0005 and a weight decay of 0.05. Similar to [52, 61], we find setting $\beta_2 = 0.98$ and $\epsilon = 10^{-6}$ helps improve stability when large-scale training. Besides, gradient clipping with 0.5 is used to stabilize training.

Uni-Perceiver-B and Uni-Perceiver-L are equipped with Conditional-MoEs layer for every other layers while Uni-Perception-Ti use Conditional MoEs in all layers. A normal noise is also used on the gate logits following [64] for a better exploration for new potential experts. If not specialized, top-2 gate function is used. For other hyper-parameters of MoE layers, please refer to Appendix.

## 4.3 Ablation Studies

This part explores whether Conditional MoEs can effectively mitigate task interference in generalist models and compares different routing strategies. Tab. 3 summarizes the performance of Uni-Perceiver and its variants on three typical tasks. Compared with task-specific parameterization, the performance degradation of Uni-Perceiver confirms the existence of task interference. Incorporating Conditional MoEs with any routing strategy can mitigate the task-interference issue and significantly improve the performance. Among these five routing strategies, token-level, context-level, and modality-level MoEs deliver slightly worse performance. We argue the missed task information is critical for resolving the task interference. Besides, the data-dependent MoEs, *i.e.*, token-level and context-level, have relatively higher training and inference cost, while the other three MoEs have excellent efficiency by using reparameterization techniques. Although both task-level and attribute MoEs achieve good performance, the specialized task-id design in task-level MoEs makes it difficult to generalize to new tasks. Therefore, the Conditional MoEs with attribute routing strategy will be used.

---

[1]As far as we know, all datasets do not contain any personally identifiable information or offensive content.

Table 4: The performance of incorporating Conditional MoEs with Uni-Perceiver on image classification, video classification and image-text retrieval. "#param" is the parameters required during model deployment. "#data" is the amount of visual training samples involved. "WT", "PT", and "FT" indicate w/o tuning, prompt tuning, and fine-tuning, respectively. "1%" and "100%" indicate the proportion of downstream data used for tuning. "$FT_{100\%}\uparrow$" means fine-tuning with larger image size. The subscript number next to score indicates that a different image resolution than 224 is used. † These methods use $> 20\times$ training data size and $> 10\times$ training cost than ours.

(a) Image Classification accuracy on ImageNet-1k.

| Method | #param | #data | WT | $PT_{1\%}$ | $FT_{100\%}$ | $FT_{100\%}\uparrow$ |
|---|---|---|---|---|---|---|
| DeiT-B [76] | 86M | 1.28M | - | - | 81.8 | $83.1_{384}$ |
| ViT-B [73] | 86M | 15.5M | - | - | 84.0 | $85.5_{384}$ |
| ViT-L [73] | 307M | 15.5M | - | - | 84.0 | $85.6_{384}$ |
| OFA [79] | 472M | 60.6M | - | - | - | $84.9_{480}$ |
| CLIP [61] | 307M | 400M | $76.2_{336}$ | - | - | - |
| †ALIGN [33] | 480M | 1.8B | $76.4_{289}$ | - | - | $88.6_{289}$ |
| †Florence [89] | 637M | 900M | $83.7_{384}$ | - | - | $90.0_{>384}$ |
| †CoCa-B [87] | 86M | 4.8B | $82.6_{576}$ | - | - | $88.3_{576}$ |
| †CoCa-L [87] | 303M | 4.8B | $84.8_{576}$ | - | - | $90.2_{576}$ |
| †Flamingo-3B [3] | 3.2B | 2.3B | - | $71.0_{320}$ | - | - |
| Uni-Perceiver-B | 86M | 44.1M | 79.2 | 80.9 | 84.0 | $85.2_{384}$ |
| + Conditional MoEs | 86M | 44.1M | 80.3 | 82.0 | 84.5 | $85.8_{384}$ |
| Uni-Perceiver-L | 354M | 303M | 82.7 | 84.2 | 86.2 | $87.0_{384}$ |
| + Conditional MoEs | 303M | 44.1M | 83.4 | 84.9 | 86.4 | $87.0_{384}$ |

(b) Video classification accuracy on Kinetics-400.

| Method | #param | #data | WT | $PT_{1\%}$ | $FT_{100\%}$ |
|---|---|---|---|---|---|
| TimeSformer-B [7] | 121.4M | 14.2M | - | - | 80.7 |
| VATT-B [2] | 87.9M | 238M | - | - | $79.6_{320}$ |
| VATT-L [2] | 306.1M | 238M | - | - | $82.1_{320}$ |
| ViViT-L [5] | >307M | 14.2M | - | - | 81.7 |
| ViViT-L [5] | >307M | 300M | - | - | 84.9 |
| †Florence [89] | 647M | 900M | - | - | $86.5_{384}$ |
| †CoCa [87] | 2.1B | 4.8B | - | - | $88.9_{576}$ |
| Uni-Perceiver-B | 86M | 44.1M | 74.5 | 74.8 | 77.7 |
| + Conditional MoEs | 86M | 44.1M | 76.8 | 77.2 | 79.3 |
| Uni-Perceiver-L | 303M | 44.1M | 79.5 | 80.0 | 81.9 |
| + Conditional MoEs | 303M | 44.1M | 82.1 | 83.0 | 84.2 |

(c) Image-text retrieval R@1 performance.

| Method | | | Flickr30K | | | | | | MSCOCO Caption | | | | | |
|---|---|---|---|---|---|---|---|---|---|---|---|---|---|---|
| | | | Image → Text | | | Text → Image | | | Image → Text | | | Text → Image | | |
| | #param | #data | WT | $PT_{1\%}$ | $FT_{100\%}$ | WT | $PT_{1\%}$ | $FT_{100\%}$ | WT | $PT_{1\%}$ | $FT_{100\%}$ | WT | $PT_{1\%}$ | $FT_{100\%}$ |
| ImageBERT [60] | 170M | 10M | 70.7 | - | 87.0 | 54.3 | - | 73.1 | 44.0 | - | 66.4 | 32.3 | - | 50.5 |
| UNITER-B [13] | 146M | 9.6M | 80.7 | - | 85.9 | 66.2 | - | 72.5 | - | - | 64.4 | - | - | 50.3 |
| UNITER-L [13] | 363M | 9.6M | 83.6 | - | 87.3 | 68.7 | - | 75.6 | - | - | 65.7 | - | - | 52.9 |
| ViLT [39] | 87M | 9.7M | 73.2 | - | 74.8 | 56.5 | - | 61.5 | 55.0 | - | 64.4 | 40.4 | - | 42.7 |
| FLAVA [71] | 215M | 70M | 67.7 | - | - | 65.2 | - | - | 42.7 | - | - | 38.4 | - | - |
| CLIP [61] | 417M | 400M | $88.0_{336}$ | - | - | $68.7_{336}$ | - | - | $58.4_{336}$ | - | - | $37.8_{336}$ | - | - |
| †ALIGN | 820M | 1.8B | $88.6_{289}$ | - | $95.3_{289}$ | $75.7_{289}$ | - | $84.9_{289}$ | $58.6_{289}$ | - | $77.0_{289}$ | $45.6_{289}$ | - | $59.9_{289}$ |
| †Florence [89] | 893M | 900M | $90.9_{384}$ | - | $97.2_{384}$ | $76.7_{384}$ | - | $87.9_{384}$ | $64.7_{384}$ | - | - | $47.2_{384}$ | - | - |
| †CoCa-B [87] | 383M | 4.8B | $89.8_{576}$ | - | - | $76.8_{576}$ | - | - | $63.8_{576}$ | - | - | $47.5_{576}$ | - | - |
| †CoCa-L [87] | 787M | 4.8B | $92.5_{576}$ | - | - | $80.4_{576}$ | - | - | $66.3_{576}$ | - | - | $51.2_{576}$ | - | - |
| †Flamingo-3B [3] | 3.2B | 2.3B | $89.3_{320}$ | - | - | $79.5_{320}$ | - | - | $65.9_{320}$ | - | - | $48.0_{320}$ | - | - |
| Uni-Perceiver-B | 124M | 44.1M | 82.3 | 91.0 | 92.7 | 71.1 | 76.0 | 77.5 | 64.9 | 68.4 | 69.8 | 50.7 | 51.9 | 53.9 |
| + Conditional MoEs | 167M | 44.1M | 82.1 | 91.3 | 93.6 | 72.4 | 78.5 | 79.8 | 64.6 | 68.9 | 70.5 | 51.6 | 52.6 | 54.1 |
| Uni-Perceiver-L | 354M | 44.1M | 83.7 | 92.1 | 94.7 | 74.2 | 80.0 | 82.1 | 67.8 | 73.3 | 74.4 | 54.1 | 56.2 | 57.9 |
| + Conditional MoEs | 505M | 44.1M | 83.6 | 92.4 | 94.1 | 75.9 | 80.6 | 83.7 | 67.9 | 73.3 | 74.7 | 55.3 | 57.1 | 58.3 |

## 4.4 Evaluation on Pre-training tasks

Large-scale training is conducted to verify the effectiveness of our method, we first evaluate it on tasks involved in pre-training. Specifically, we use widely-used Imagenet-1k [18] and Kinetics-400 [37] to evaluate image and video classification respectively, and use popular Flickr30k [59] and COCO Caption [12] to evaluate image caption and image-text retrieval.

Tab. 4 and Tab. 5a show the results on the four pre-training tasks. We see that Uni-perceiver with our Conditional MoEs consistently outperforms vanilla Uni-perceiver by a large margin. Without any tuning, our models achieve comparable performance with task-specific SOTAs trained with similar model size and training data size. Note that, our approach is a generalist model pretrained on a unified task formulation, while task-specific approaches are trained specifically for the target task.

When prompt tuned on only 1% downstream data, the performance of our models are boosted to a level close to counterparts that use $>50\times$ training data sizes and $>10\times$ training cost. For the prompt tuning of our models, only a small amount of parameters are tuned, and the encoder is still fixed and shared among different tasks, indicating that generalist models with Conditional MoEs can handle different tasks with significant low cost than counterparts.

We further fine-tune our models with 100% of the downstream data. In this case, our model achieves performance on-par with or better than the SOTAs trained with similar data size on all these tasks, which proves generalist models with Conditional MoEs has learned high-quality representations.

Table 5: The performance of incorporating Conditional MoEs with Uni-Perceiver on image caption, natural language understanding, video-text retrieval and video caption, where the last three tasks are not involved in pre-training.

(a) Image caption BLEU@4 performance. * means methods use region features as network inputs. ‡ indicates that Cider optimization is used.

| Method | #param | data | MSCOCO Caption | | | Flickr30k | | |
|---|---|---|---|---|---|---|---|---|
| | | | WT | $PT_{1\%}$ | $FT_{100\%}$ | WT | $PT_{1\%}$ | $FT_{100\%}$ |
| *Unified VLP [92] | 86M | 3.0M | - | - | 36.5 | - | - | 30.1 |
| *OSCAR-B [46] | 154M | 6.5M | - | - | 36.5 | - | - | - |
| *OSCAR-L [46] | 384M | 6.5M | - | - | 37.4 | - | - | - |
| UNICORN [86] | 198M | 200k | - | - | 35.8 | - | - | - |
| BLIP-B [44] | 252M | 129M | - | - | $39.7_{384}$ | - | - | - |
| BLIP-L [44] | 473M | 129M | - | - | $40.4_{384}$ | - | - | - |
| CLIP-VIL [69] | >459M | 400M | - | - | 40.2 | - | - | - |
| SimVLM [84] | 632M | 1.8B | $11.2_{480}$ | - | $40.6_{480}$ | - | - | - |
| OFA [79] | 472M | 60.6M | - | - | $42.4_{480}$ | - | - | - |
| *‡OSCAR-L [46] | 384M | 6.5M | - | - | 41.7 | - | - | - |
| †CoCa [87] | 2.1B | 4.8B | | | $40.9_{576}$ | - | - | - |
| Uni-Perceiver-B | 124M | 44.1M | 32.0 | 35.5 | 36.4 | 14.7 | 30.2 | 31.2 |
| + Conditional MoEs | 167M | 44.1M | 33.2 | 36.8 | 37.3 | 15.9 | 30.7 | 32.4 |
| Uni-Perceiver-L | 354M | 44.1M | 35.3 | 38.6 | 39.2 | 15.1 | 32.9 | 35.5 |
| + Conditional MoEs | 505M | 44.1M | 35.5 | 39.3 | 40.5 | 15.8 | 33.7 | 36.2 |

(b) Natural language understanding (novel task) fine-tuned on GLUE. $BERT_{BASE}$ records from [34]. VisualBERT and LXMERT record from [30]. *RoBERTa uses $10\times$ training text tokens than ours.

| Method | MNLI (Acc) | QNLI (Acc) | QQP (F1) | RTE (Acc) | SST-2 (Acc) | MRPC (F1) | CoLA (Mcc) |
|---|---|---|---|---|---|---|---|
| LXMERT [75] | 80.4 | 84.2 | 75.3 | 57.2 | 90.2 | 80.4 | 39.0 |
| VisualBERT [45] | 81.6 | 87.0 | 86.0 | 56.6 | 89.4 | 82.1 | 38.6 |
| SimVLM-B [84] | 83.4 | 88.6 | 87.2 | 63.9 | 90.9 | 84.4 | 46.7 |
| BERT-B [78] | 84.5 | 88.4 | 88.3 | 63.5 | 92.9 | 89.0 | 54.7 |
| BERT-L [78] | 86.6 | 92.3 | 91.3 | 70.4 | 93.2 | 88.0 | 60.6 |
| OFA-B [79] | 84.3 | 91.1 | 88.4 | 70.8 | 92.7 | 90.6 | 52.3 |
| OFA-L [79] | 86.6 | 92.8 | 88.9 | 73.6 | 94.7 | 91.4 | 53.1 |
| *RoBERTa-B [52] | 87.6 | 92.8 | 91.9 | 78.7 | 94.8 | 90.2 | 63.6 |
| *RoBERTa-L [52] | 90.2 | 94.7 | 92.2 | 86.6 | 96.4 | 90.9 | 68.0 |
| Uni-Perceiver-B | 79.7 | 87.3 | 86.7 | 71.1 | 89.3 | 86.0 | 43.1 |
| + Conditional MoEs | 81.5 | 88.2 | 87.8 | 75.8 | 90.9 | 87.1 | 52.2 |
| Uni-Perceiver-L | 82.5 | 89.2 | 87.7 | 73.7 | 91.2 | 90.2 | 52.0 |
| + Conditional MoEs | 85.7 | 91.9 | 89.5 | 78.4 | 93.4 | 91.2 | 57.4 |

(c) Video-text retrieval (novel task) Recall@1 and video caption (novel task) BLEU@4 performance on MSVD.

| Method | #param | #data | Video $\rightarrow$ Text | | | Text $\rightarrow$ Video | | | Video Caption | | |
|---|---|---|---|---|---|---|---|---|---|---|---|
| | | | WT | $PT_{1\%}$ | $FT_{100\%}$ | WT | $PT_{1\%}$ | $FT_{100\%}$ | WT | $PT_{1\%}$ | $FT_{100\%}$ |
| CLIP2video [22] | 132M | 400M | - | - | 58.7 | - | - | 47.0 | - | - | - |
| HunYuan_tvr [56] | 364M | 400M | - | - | 68.0 | - | - | 52.7 | - | - | - |
| ORG-TRL [91] | 86M | 2.0M | - | - | - | - | - | - | - | - | 54.3 |
| Uni-Perceiver-B | 124M | 44.1M | 50.3 | 62.7 | 62.8 | 38.7 | 43.8 | 45.8 | 22.6 | 59.5 | 63.3 |
| + Conditional MoEs | 167M | 44.1M | 52.8 | 65.6 | 65.0 | 40.0 | 45.3 | 47.8 | 23.4 | 60.0 | 65.4 |
| Uni-Perceiver-L | 354M | 44.1M | 45.4 | 65.5 | 65.2 | 34.2 | 48.6 | 50.8 | 24.7 | 67.2 | 68.3 |
| + Conditional MoEs | 505M | 44.1M | 45.7 | 66.4 | 67.6 | 41.9 | 50.3 | 52.3 | 24.6 | 67.6 | 68.9 |

## 4.5 Generalization to Novel Tasks

The generalization ability is the most attractive aspect of generalist models, while the dynamic sub-networks activation of Conditional MoEs should maintain this ability while mitigating task interference. To verify this, we conduct experiments on video caption, video-text retrieval, and natural language understanding tasks, which did not appear in pre-training. As shown in Tab. 5c, our Uni-Perceiver equipped with Conditional MoEs could generalize to video-related tasks very well. They can obtain reasonable zero-shot performance on those tasks and also perform better than vanilla Uni-Perceiver with a great margin. Moreover, Uni-Perceiver-MoEs can achieve comparable results to SOTA methods with similar training cost by further conducting prompt tuning with only 1% data. Beyond that, Conditional MoEs can significantly boost the performance of Uni-Perceiver on GLUE benchmarks (Tab. 5b), owing to its excellent ability to resolve task interference in generalist models.

## 5 Conclusion

In this paper, we propose Conditional MoEs to address the task-interference issue in generalist models. By sparsely activate sub-networks without introducing any task-specific designs, generalist models can be pre-trained on multiple tasks jointly without performance degradation, while keeping the generalization ablity to novel tasks. We incorporate Conditional MoEs with the recently proposed generalist model Uni-Perceiver. With prompt tuning on 1% downstream data, the proposed sparse generalist model achieves competitive performance with previous SOTAs using only <5% training data and <10% training cost. We hope this work can motivate further research in generalist models.

**Limitations.** Our method is currently verified on generalist models with millions of parameters. For generalist models with billions of parameters, whether the task-interference issue exists and whether our method is still effective are questionable, which we leave them to future work.

**Potential Negative Societal Impact.** This work shares the common negative impacts of large-scale training, which may consume lots of electricity and result in increased carbon emissions. This method also learns from a large number of datasets that may contain data biases.

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
