Table 6: Uni-Perceiver model variants used in this paper. "#vocab params" and "#encoder params" represent the number of vocabulary parameters and encoder network parameters, respectively.

| | base model | embedding dimension | #heads | #layers | #vocab params | #encoder params | #total params |
|---|---|---|---|---|---|---|---|
| Uni-Perceiver-Ti | DeiT-Ti [76] | 192 | 3 | 12 | 9.5M | 5.3M | 14.8M |
| Uni-Perceiver-B | ViT-B [20] | 768 | 12 | 12 | 38M | 86M | 124M |
| Uni-Perceiver-L | ViT-L [20] | 1024 | 16 | 24 | 51M | 303M | 354M |

Table 7: Tasks and datasets used for our pre-training.

| task | dataset | batch size/GPU | sampling weight | loss weight |
|---|---|---|---|---|
| Image Classification | ImageNet-21k [18] | 220 | 0.2486 | 1.0 |
| Video Classification | Kinetics-700 [37] | 6 | 0.01 | 0.1 |
| | Moments in Time [57] | 24 | 0.02 | 0.1 |
| Masked Language Modeling | Books&Wiki [94] | 256 | 0.275 | 0.5 |
| Image Caption | YFCC [35] | 100 | 0.0584 | 1.0 |
| | CC12M [9] | 100 | 0.05057 | 1.0 |
| | CC3M [66] | 100 | 0.026295 | 1.0 |
| | Visual Genome [41] | 100 | 0.01766 | 1.0 |
| | COCO Caption [12] | 100 | 0.01144 | 1.0 |
| | SBU [58] | 100 | 0.01383 | 1.0 |
| Image-Text Retrieval | YFCC [35] | 160 | 0.0584 | 0.5 |
| | CC12M [9] | 160 | 05057 | 0.5 |
| | CC3M [66] | 160 | 0.026295 | 0.5 |
| | Visual Genome [41] | 160 | 0.01766 | 0.5 |
| | COCO Caption [12] | 160 | 0.01144 | 0.5 |
| | SBU [58] | 160 | 0.01383 | 0.5 |

# 6 Appendix

## 6.1 Experimental Details

**Pre-training Details.** Uni-Perceivers with three variants are used in our works, which are summarized in Tab. 6. Uni-Perceiver-Ti adopts the same setting of DeiT-Ti [76] in ablation experiments. Uni-Perceiver-B and Uni-Perceiver-L have the same architectures as their corresponding ViT variants, respectively. We follow most of the settings in Uni-Perceiver [93]: cross-entropy loss with label smoothing of 0.1 is adopted for all tasks, and the negative samples for retrieval tasks are only from the local batch in the current GPU. We also apply the same data augmentation techniques as Uni-Perceiver [93] to image and video modalities to avoid overfitting. The Uni-Perceiver and Uni-Perceiver-MoE models are pre-trained on 32 NVIDIA-A100 GPUs (80GB memory) for 400k iterations.

There are some setting changes to improve the training stability of the original Uni-Perceiver. Following [102], a uniform drop rate for stochastic depth is used across all encoder layers and are adapted according to the model size. Additionally, LayerScale [101] is used to facilitate the convergence of Transformer training, and the same initialization of $10^{-3}$ is set to all models for simplicity. Besides, Tab. 7 lists the batch size, sampling weight, and loss weight for each task and dataset in the pre-training stage. The loss weights are adjusted to meet reasonable optimizations for all tasks by observing the early training losses through short-epoch experiments. Those sampling weights for each task and dataset are proportional to the square root of the dataset size, which is demonstrated to be an effective heuristic to ease data imbalance across different datasets [4]. Following [102], we first pre-train with the image resolution of $160 \times 160$ and the patch size of $16 \times 16$, and continue pre-training for another 10% of total iterations on a higher resolution of $224 \times 224$. Furthermore, the implementation of mixed precision training in [100] is also employed to train Uni-Perceiver with a larger batch size. Based on the above settings, we can train Uni-Perceiver more efficiently. As shown in Tab. 8, our re-implemented Uni-Perceiver also achieves better performance across various tasks.

The number of experts in each Conditional-MoE layer is set to 8 by default. For Conditional MoEs with data-dependent routing strategies, *i.e.,* token-level and context-level, we use a capacity factor of 1.0 for the training stage and 2.0 for the evaluation stage [23, 40]. Besides, The balance loss in [23] is also employed for data-dependent MoEs to accomplish the balanced load of expert utilization, which is added to the total loss with a multiplicative coefficient of $10^{-2}$.

Table 8: The comparison between the original Uni-Perceiver-B [93] and our re-implemented Uni-Perceiver-B[*]. Results are reported for image classification accuracy on ImageNet-1K, video classification accuracy on Kinetics-400, image-text retrieval R@1 on Flickr30K, and image caption BLEU@4 on COCO Caption. 'TR' and 'IR' represent text retrieval and image retrieval, respectively.

| | ImageNet-1k | | | Kinetics-400 | | | Flickr30K | | COCO Caption | pre-training time |
| | WT | $PT_{1\%}$ | $FT_{100\%}$ | WT | $PT_{1\%}$ | $FT_{100\%}$ | $TR(FT_{100\%})$ | $IR(FT_{100\%})$ | $FT_{100\%}$ | TPU-v3-core-days |
|---|---|---|---|---|---|---|---|---|---|---|
| Uni-Perceiver-B [93] | 78.0 | 80.2 | 83.8 | 73.5 | 73.6 | 75.8 | 87.9 | 74.9 | 35.6 | $\sim 3.4$k |
| Uni-Perceiver-B[*] | 79.2 | 80.9 | 84.0 | 74.5 | 74.8 | 77.7 | 92.7 | 77.5 | 36.4 | $\sim 1.0$k |

Table 9: Hyper-parameters for tuning on the downstream tasks. "*param1/param2*" denotes the corresponding parameters used for fine-tuning and prompt tuning, respectively.

| | ImageNet-1k | Kinetics-400 | COCO Caption | Flickr30K Caption | COCO Retrieval | Flickr30K Retrieval |
|---|---|---|---|---|---|---|
| Gradient clip | | | 1.0/1.0 | | | |
| Stoch. Depth | | | 0.2/0.2 | | | |
| Weight decay rate | | | $1\times10^{-4}$/0.0 | | | |
| LR decay schedule | | | Cosine Schedule Decaying to Zero/Constant Learning Rate | | | |
| Train steps | 20k/5k | 20k/2k | 10k/1k | 4k/200 | 10k/500 | 5k/100 |
| Train batch size | 2048/1024 | 64/32 | 512/512 | 512/512 | 2048/2048 | 2048/2048 |
| Warm-up steps | 2k/500 | 2k/200 | 1k/100 | 400/20 | 1k/50 | 500/10 |
| Learning rate | $2\times10^{-5}/1\times10^{-3}$ | $5\times10^{-6}/5\times10^{-4}$ | $2\times10^{-5}/1\times10^{-3}$ | $2\times10^{-5}/1\times10^{-3}$ | $5\times10^{-6}/1\times10^{-3}$ | $5\times10^{-6}/1\times10^{-3}$ |

**Fine-tuning & Prompt Tuning.** In addition to evaluation without any parameter tuning, fine-tuning with 100% data and prompt tuning with 1% data are also conducted to evaluate the model performance. We mainly follow the practice in Uni-Perceiver. Specifically, for prompt tuning, following P-Tuning v2 [97], learnable prompt tokens with random initialization are added at each encoder layer, and class labels with linear heads are added for classification tasks. The <SPE> token and layer norm parameters are also tuned. All training receipts for fine-tuning and prompt tuning are listed in Tab. 9.

**Removing Overlap.** Following [93], we carefully remove those videos overlapping with the validation set of Kinetics-400 in the training set of Kinetics-700.

## 6.2 The Placement of Conditional MoEs

Previous methods usually only incorporate the MoE layers with every other dense feed-forward network (FFN) layer [23, 40]. Conversely, we prefer using Conditional-MoE layers in every layer in the transformer, which is beneficial for mitigating task interference thoroughly. Tab. 11 shows the results when applying Conditional-MoE layers at intervals or only on certain type of layers, *i.e.,* FFN layers or self-attention layers. We can observe that the more layers are replaced with attribute-level MoEs, the more the task interference will be mitigated and the higher performance of each task can be achieved. Besides, applying Conditional MoEs in both self-attention and FFN layers can better alleviate the task interference. Nevertheless, only equipping self-attention layers with Conditional MoEs has limited ability to resolve the task interference.

## 6.3 Visualization of Gating decisions for Attribute-level MoEs

We show the expert distributions of different layers of a trained Uni-Perceiver-MoE model in Fig. 3. Both Conditional MoEs in self-attention layers (Fig. 3a) and FFN layers (Fig. 3b) have learnt to activate experts sparsely according to the token attributes. There are some experts shared by tokens with the same modality. For example, "images" for image classification and image caption tasks are usually routed to the same experts. Additionally, there are also some experts shared by the inputs and targets from the same task, *e.g.,* the expert *b* in the 7*th* FFN layer. Interestingly, the attribute of causal attention mask is also utilized by Conditional MoEs, *e.g.,* the 2*nd* and the 9*th* self-attention layers, indicating the potential interference between auto-encoding and auto-regressive tasks.

## 6.4 Training Statistics of Relevant Methods

Tab. 10 lists the training statistics of some methods relevant to our work. As discussed in Sec. 2, these methods are divided into three categories: specialized models, integrated specialized models, and generalist models. For a fair comparison, all of the data used to train each method from scratch is recorded, including the data used in the off-the-shelf pre-trained models. We can see that the training data scale of Uni-Perceiver-MoE is relatively much smaller than most of the generalist models and comparable with some specialized models.

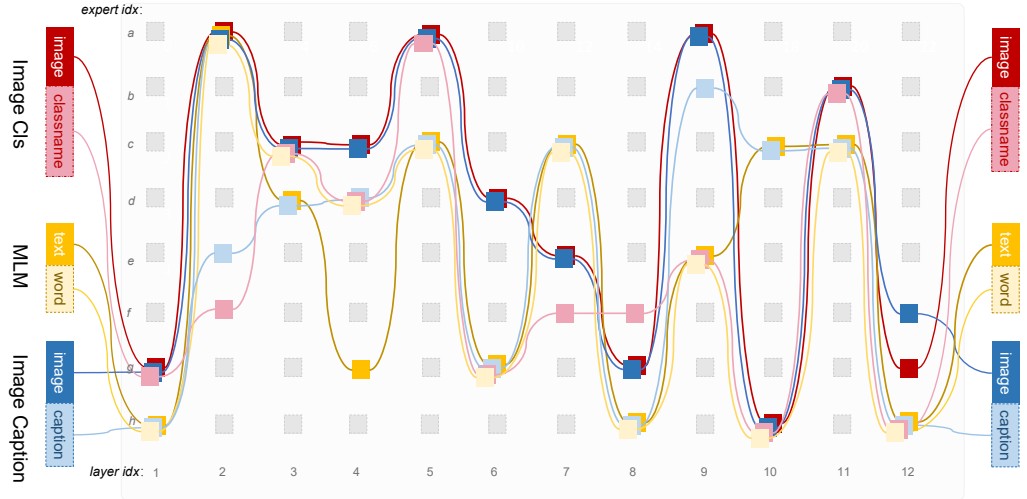

(a) Gating decisions of the self-attention layers for Uni-Perceiver-MoE-Ti.

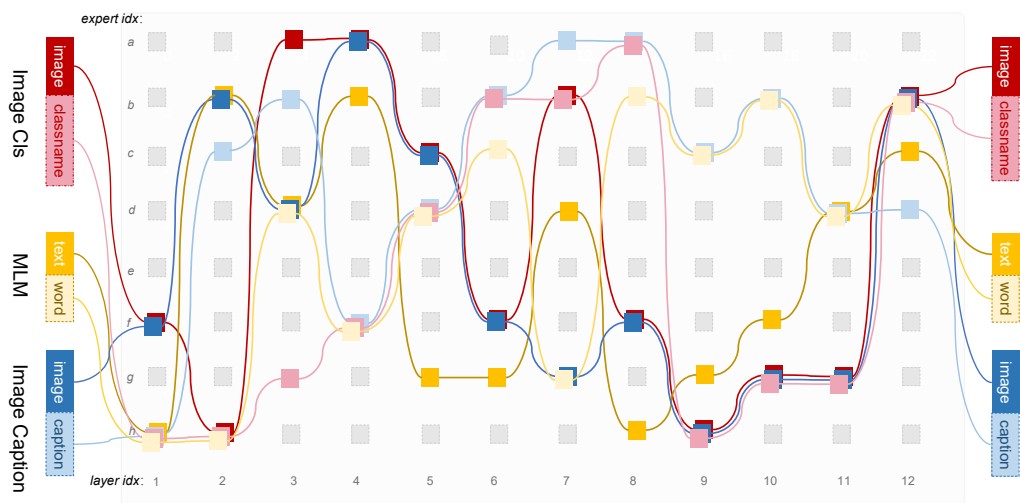

(b) Gating decisions of the FFN layers for Uni-Perceiver-MoE-Ti.

Figure 3: Gatinng decisions of the trained Uni-Perceiver-Ti with attribute-level MoEs. We show the top-1 activation of experts in every self-attention and FFN layer on three tasks: image classification (Image Cls) on ImageNet-1k, Masked Language Modeling (MLM) on Books&Wiki, and image caption on COCO Caption. The corresponding activation for the input and target from different tasks are highlighted with different colors. Grey shaded squares represent those experts that are not activated.

Table 10: Training statistics of relevant methods, which are categorized into specialized models, integrated specialized models, and generalist models. * represents the storage space of the plain texts. † represents the number of tokens in RL tasks. ¶ indicates the data used in the off-the-shelf pre-trained models. § datasets are private.

| methods | training data | | | total visual data size | training time TPU-v3-core-days[1] |
|---|---|---|---|---|---|
| | dataset | data type | data size | | |
| ▶ *Specialized Models* | | | | | |
| BERT [19] | Books&Wiki | plain texts | 16GB | - | 1.4K |
| ORG-TRL [91] | ImageNet-1K | images | 1.28M¶ | 1.6M | - |
| | Kinetics-400 | videos | 300K¶ | | |
| | MSCOCO | image-bounding boxes | 118K¶ | | |
| | Books&Wiki | plain texts | 16GB*¶ | | |
| Unified VLP [92] | CC3M | image-text pairs | 3.0M | 3.1M | - |
| | VG | image-bounding boxes | 108K¶ | | |
| | Books&Wiki | plain texts | 16GB*¶ | | |
| OSCAR [46] | COCO, CC3M, SBU, Flickr30K, VQA, GQA, VG-QA | image-text pair | 6.5M | 6.6M | - |
| | VG | image-bounding boxes+attributes | 108K¶ | | |
| UNITER[13] | CC3M, SBU, COCO, VG | image-text pair | 9.6M | 9.7M | 0.5K |
| | VG | image-bounding boxes+attributes | 108K¶ | | |
| ImageBERT [60] | CC3M, SBU, LAIT§ | image-text pairs | 13.7M | 13.8M | - |
| | VG | image-bounding boxes | 108K¶ | | |
| | Books&Wiki | plain texts | 16GB¶ | | |
| TimeSformer [7] | ImageNet-21K | images | 14.2M¶ | 14.5M | - |
| | Kinetics-400 | videos | 300K | | |
| ViT-L [73] | ImageNet-21K, ImageNet-1K | images | 15.5M | 15.5M | 0.23K |
| ViLT [39] | COCO, VG, CC3M, SBU | image-text pairs | 9.7M | 25.2M | - |
| | ImageNet-21K, ImageNet-1K | images | 15.5M¶ | | |
| VATT [2] | AudioSet | audios | 2.1M | 138M | 1.5K |
| | HowTo100M | video-audio-text triplets | 136M | | |
| BLIP [44] | COCO, VG, SBU CC3M, CC12M, LAION | image-text pairs | 130M | 144M | - |
| | ImageNet-21k | images | 14.2M¶ | | |
| | Books&Wiki | plain texts | 16GB*¶ | | |
| ViViT [5] | JFT-300M§, ImageNet-21K | images | 300M¶ | 300.3M | - |
| | Kinetics-400 | videos | 300k | | |
| CLIP [61] | CLIP data§ | image-text pairs | 400M | 400M | 18K |
| HunYuan_tvr [56] | CLIP data§ | image-text pairs | 400M | 400M | - |
| CLIP2video [22] | CLIP data§ | image-text pairs | 400M | 400M | - |
| CLIP-VIL [69] | CLIP data§ | image-text pairs | 400M | 400M | - |
| ALIGN [33] | ALIGN data§ | image-text pairs | 1.8B | 1.8B | - |
| ▶ *Integrated Specialized Models* | | | | | |
| FLAVA [71] | COCO, SBU, Localized Narratives, CC3M, VG, Wikipedia Image Text, CC12M, Red Caps, YFCC | image-text pairs | 70M | 71.3M | - |
| | ImageNet-1K | images | 1.28M | | |
| | CCNews, BookCorpus | plain texts | 970GB* | | |
| Florence [89] | FLB-900M§ | image-text pairs | 900M | 909M | 44K |
| | FLOD-9M§ | image-bounding boxes | 9M | | |
| ▶ *Generalist Models* | | | | | |
| UNICORN [86] | COCO, VG, CC3M, SBU | image-text pairs | 9.8M | 25.3M | - |
| | ImageNet-21K, ImageNet-1K | images | 15.5M¶ | | |
| | BooksCorpus, CC-News, OpenwebText, Stories | plain texts | 160GB¶ | | |
| OFA [79] | CC12M, CC3M, SBU, COCO, VG-Cap | image-text pairs | 15.25M | 60.6M | - |
| | VQAv2, VG-QA, GQA | visual question answering | 2.92M | | |
| | RefCOCO, RefCOCO+, RefCOCOg, VG-Cap | image-instance-text triplets | 3.2M | | |
| | OpenImages, Object365, VG, COCO | image-bounding boxes | 3.0M | | |
| | YFCC100M, ImageNet-21K | Images | 36.27M | | |
| | Pile | plain texts | 140GB* | | |
| SimVLM [84] | ALIGN data§ | image-text pairs | 1.8B | 1.8B | - |
| | Colossal Clean Crawled Corpus | plain texts | 800GB* | | |
| Gato [62] | DM Lab, ALE Atari, ALE Atari Extended, Sokoban, BabyAI, DM Control Suite, Procgen Benchmark, RGB Stacking simulator, RGB Stacking real robot, Meta-World, DM Manipulation Playground, Playroom | simulated data for RL tasks | 1.5T† | 2.16B | 2K |
| | M3W§, ALIGN data§, CC3M, COCO LTIP§, QKVQA, VQAV2 | image-text pairs | 2.16B | | |
| | MassiveWeb | plain texts | 1.9TB* | | |
| Flamingo [3] | M3W§, ALIGN data§, LTIP§ | image-text pairs | 2.3B | 2.3B | 126K |
| | VTP§ | video-text pair | 27M | | |
| CoCa [87] | JFT-3B§, ALIGN data§ | image-text pairs | 4.8B | 4.8B | 56K |
| Uni-Perceiver [93] & Uni-Perceiver-MoE | CC3M, CC12M, SBU, COCO, VG, YFCC | image-text pairs | 28.6M | 44.1M | 4.2K |
| | ImageNet-21K | images | 14.2M | | |
| | Kinetics-700, Moments in Time | videos | 1.33M | | |
| | Books&Wiki | plain texts | 16GB* | | |

---

[1]We convert the training time to the TPU-v3-core-days based on the TFLOPS of accelerators, *i.e.,* 1.0 TPU-v3-core-days ≈ 0.364 TPU-v4-core-days ≈ 0.151 NVIDIA-A100-days ≈ 0.302 NVIDIA-V100-days.

Table 11: The performance of different settings of attribute-level routings. FFN MoE and SA MoE indicate whether Conditional MoEs are applied in FFN layers and self-attention layers, respectively. every-$n$ means Conditional-MoE layers are placed every $n$ layers in the Transformer encoder.

| model | FFN MoE | SA MoE | every-$n$ | ImageNet-1k | | COCO Caption | | MLM | |
|---|---|---|---|---|---|---|---|---|---|
| | | | | $\uparrow$acc$_{train}$ | $\uparrow$acc$_{val}$ | $\uparrow$acc$_{train}$ | $\uparrow$B@4$_{val}$ | $\uparrow$acc$_{train}$ | $\downarrow$ppl$_{val}$ |
| Uni-Perceiver-Ti | - | - | - | 47.3 | 68.3 | 49.2 | 18.2 | 54.5 | 5.86 |
| + Conditional MoEs $_{attribute}$ | ✓ | ✓ | 4 | 51.7 | 71.6 | 51.3 | 20.9 | 56.1 | 5.47 |
| + Conditional MoEs $_{attribute}$ | ✓ | ✓ | 2 | 52.2 | 72.3 | 51.8 | 21.1 | 57.3 | 5.13 |
| + Conditional MoEs $_{attribute}$ | ✓ | ✗ | 1 | 51.5 | 71.3 | 52.2 | 21.0 | 58.5 | 4.86 |
| + Conditional MoEs $_{attribute}$ | ✗ | ✓ | 1 | 49.2 | 69.7 | 51.0 | 20.6 | 55.8 | 5.50 |
| + Conditional MoEs $_{attribute}$ | ✓ | ✓ | 1 | 52.8 | 73.3 | 53.1 | 23.0 | 60.0 | 4.56 |

## 6.5 Licenses of Datasets

**ImageNet-21K** [18] is subject to the ImageNet terms of use [103].

**Kinetics-700** & **Kinetics-400** [37] The kinetics dataset is licensed by Google Inc. under a Creative Commons Attribution 4.0 International License.

**BooksCorpus** [94] Replicate Toronto BookCorpus is open-source and licensed under GNU GPL, Version 3.

**Wikipedia** Most of Wikipedia's text is co-licensed under the Creative Commons Attribution-ShareAlike 3.0 Unported License (CC BY-SA) and the GNU Free Documentation License (GFDL) (unversioned, with no invariant sections, front-cover texts, or back-cover texts). Some text has been imported only under CC BY-SA and CC BY-SA-compatible license and cannot be reused under GFDL.

**YFCC** [35] All the photos and videos provided in YFCC dataset are licensed under one of the Creative Commons copyright licenses.

**CC12M** [9] is licensed under the Terms of Use of Conceptual 12M [98].

**CC3M** [66] is licensed under the Conceptual Captions Terms of Use [99].

**Visual Genome** [41] is licensed under a Creative Commons Attribution 4.0 International License [96].

**COCO Caption** [12] The images are subject to the Flickr terms of use [95].

**SBU Caption** [58] The images are subject to the Flickr terms of use [95].

## Appendix References

[95] I. Flickr. Flickr terms & conditions of use. https://www.flickr.com/help/terms.

[96] R. Krishna. Visual genome terms & conditions of use. https://visualgenome.org/about.

[97] X. Liu, K. Ji, Y. Fu, Z. Du, Z. Yang, and J. Tang. P-tuning v2: Prompt tuning can be comparable to fine-tuning universally across scales and tasks. *arXiv preprint arXiv:2110.07602*, 2021.

[98] G. LLC. Conceptual 12m terms & conditions of use. https://github.com/google-research-datasets/conceptual-12m/blob/main/LICENSE, .

[99] G. LLC. Conceptual captions terms & conditions of use. https://github.com/google-research-datasets/conceptual-captions/blob/master/LICENSE, .

[100] J. Rasley, S. Rajbhandari, O. Ruwase, and Y. He. Deepspeed: System optimizations enable training deep learning models with over 100 billion parameters. In *Proceedings of the 26th ACM SIGKDD International Conference on Knowledge Discovery & Data Mining*, 2020.

[101] H. Touvron, M. Cord, A. Sablayrolles, G. Synnaeve, and H. Jégou. Going deeper with image transformers. *arXiv preprint arXiv:2103.17239*, 2021.

[102] H. Touvron, M. Cord, and H. Jégou. Deit iii: Revenge of the vit. *arXiv preprint arXiv:2204.07118*, 2022.

[103] P. University and S. University. Imagenet terms & conditions of use. https://image-net.org/download.