# OpenReview forum: "Uni-Perceiver-MoE: Learning Sparse Generalist Models with Conditional MoEs"
_NeurIPS.cc/2022/Conference — NeurIPS 2022 Accept_

### Official Review · Reviewer_KJqa · 2022-07-09

**Rating:** 4
**Confidence:** 3
**Soundness:** 2 fair
**Presentation:** 2 fair
**Contribution:** 2 fair

**Summary:**

This paper proposes to augment the previously proposed Uni-Perceiver model with mixture-of-experts (MoE) routing so that parameters of a model trained on several tasks can specialize by task and data modality. This choice is motivated by appealing to the phenomenon of task-interference, where the contributions to the gradient updates from different tasks for shared parameters do not perfectly align. The paper explores several ways to parameterize the MoE routing, and settles on a strategy that learns to route based on a featurization that encompasses task, token, and modality type. The resulting architecture is evaluated against ablations and strong baselines on several image, video, and text domains.

**Questions:**

- Does interference (as discussed in Table 1) correlate with task performance? It’s unclear that gradients of different orientations between tasks is evidence of difficulty optimizing their average, as claimed in L34-36.

- Does the proposed MoE block reduce the average interference metric discussed in Table 1 and used to motivate the proposed system? This is used to motivate and justify the claims at several places in the text (e.g. abstract, intro, section 2, sections 4.5), but no evidence is presented for changes to this effect.

- The MoE routing relies on a hand-designed featurization that appears to be unlikely to scale beyond a small number of tasks and datasets (as opposed to the token-level routing, which is very general). How does token-level routing perform outside of the ablation study shown in Table 3?

- The paper does not mention the Perceiver architecture by name, but only to the subsequent Uni-Perceiver architecture. This is historically misleading.

- L298: “we propose Conditional” -> “we propose Conditional MoEs”

- Footnote under L258: "konw" -> "know"

**Strengths And Weaknesses:**

**Strengths**

- The proposed MoE input featurization appears to typically improve performance when plugged into the Uni-Perceiver model.

- The MoE strategy used is more flexible than strict per-task weight sharing, while relying on similar amounts of knowledge about the input domains and tasks.

- Appendix Table 10 provides a very useful overview of the training protocols of a variety of specialist and generalist models.

**Weaknesses**

- It’s unclear that task interference (as characterized in section 3.1) either limits performance by generalist models or (if it does) that it’s addressed by the proposed MoE mechanism. Task interference is central to the claims and justification of the model, so evidence that this effect is a limiting factor is essential. See **Questions** below for more details.

- The results in Tables 4 and 5 are very hard to interpret, as they contain dozens of numbers with no clear evidence of overall improvement and no comparisons at matched performance in terms of wall-clock training, FLOPs, or other computation or memory measures.

- The proposed model doesn’t appear to strongly outperform any of the baseline models or open up new applications domains (it is applied only on standard tasks that are well-addressed by specialist methods or by other generalist methods that do not appear to require solutions to the proposed task-interference problem). What is the use-case for this model?

---

> ### Author Response · Authors · 2022-08-02
> **Reply to Reviewer KJqa (1/2)**
>
> Thank you for your review.  We will consider each of your concerns below.
>
> ---
>
> >__Q1: Does interference correlate with task performance? Do gradients of different orientations between tasks increase the difficulty of optimization？__
>
> Optimizing generalist models follows the __multi-task learning (MTL)  paradigm__, in which numerous studies have found that task interference caused by gradient conflict is the main reason for performance degradation [1,2,3,4,5]. Multi-task learning aims to tackle multiple different tasks and involves multiple objectives. It usually optimizes the weighted sum of loss across different tasks. However, those gradients from different tasks may have conflicting directions [4]. Such gradient conflict across different tasks can be detrimental to specific tasks' performance, often resulting in a much worse final performance for each task than learning them independently [5].
>
> Table 3 shows that vanilla generalist models like Uni-Perceiver also consistently perform poorly when training multiple tasks jointly, which shares the similar phenomenon as in [5]. Vanilla generalist models share all parameters among all tasks which can easily lead to the task interference phenomenon described above. In Table 1, we notice that Uni-Perceiver indeed has gradient conflicts as mentioned in [4]. Therefore, to improve generalist models' performance, it is necessary to find solutions to mitigate task interference caused by conflicting gradients.
>
> [1] Sener, Ozan, and Vladlen Koltun. "Multi-task learning as multi-objective optimization." Advances in neural information processing systems 31 (2018).
>
> [2] Crawshaw, Michael. "Multi-task learning with deep neural networks: A survey." arXiv preprint arXiv:2009.09796 (2020).
>
> [3] Yu, Tianhe, et al. "Gradient surgery for multi-task learning." Advances in Neural Information Processing Systems 33 (2020): 5824-5836.
>
> [4] Liu, Bo, et al. "Conflict-averse gradient descent for multi-task learning." Advances in Neural Information Processing Systems 34 (2021): 18878-18890.
>
> [5] Navon, Aviv, et al. "Multi-task learning as a bargaining game." arXiv preprint arXiv:2202.01017 (2022).
>
> ---
>
> >__Q2: Does the proposed MoE block reduce the average interference metric?__
>
> In fact, when different tasks are dispatched to completely different MoE experts, their gradients are orthogonal, and their corresponding average task interference metrics are naturally zero.
>
> To further illustrate the effectiveness of Conditional MoEs, we calculate the average interference metric of Uni-Perceiver-Ti with attribute MoE (whose task performance is reported in Table 3). The metric between 3 tasks is shown below.
>
>
> | The 4-th FFN Block | ImgCLS(img) | MLM (Text) | Caption(Img-Text) |
> |--------------------|:-----------:|:----------:|:-----------------:|
> | ImgCLS(img)        |     1.00    |    -0.01   |        1.57       |
> | MLM (Text)         |     0.00    |    1.00    |        0.12       |
> | Caption(Img-Text)  |     0.00    |    0.00    |        1.00       |
>
> | The 12-nd FFN Block | ImgCLS(img) | MLM (Text) | Caption(Img-Text) |
> |---------------------|:-----------:|:----------:|:-----------------:|
> | ImgCLS(img)         |     1.00    |    -0.08   |        0.20       |
> | MLM (Text)          |    -0.02    |    1.00    |        0.76       |
> | Caption(Img-Text)   |     0.00    |    0.01    |        1.00       |
>
> Compared with those results in Table 1, large negative values no longer exist now, which means
>  that task interference is reduced remarkably. Meanwhile, there are also some large positive values in the current metrics, which means that Conditional MoEs allow task collaboration.
>
> ---
>
> > __Q3: The results in Tables 4 and 5 are very hard to understand.__
>
> We are sorry for the confusion. We have reorganized the results of Table 4 and Table 5 in Appendix Table 12 and Table 13, respectively, which can also be viewed at  https://ibb.co/2g8CrzR and https://ibb.co/3B3HwZF. The compared methods are divided into three groups. The 1st and 2nd groups contain those methods with similar model parameters and training data as Uni-Perceiver-Base and Uni-Perceiver-Large, respectively. The 3rd group contain other SOTAs that cannot be directly comparable to our methods, e.g., methods using far more data or training cost than ours (usually >10 times), or using extra optimization techniques (e.g., cider optimization).
>
>
> ---

---

> > ### Author Response · Authors · 2022-08-02
> > **Reply to Reviewer KJqa (2/2)**
> >
> > ---
> >
> > > __Q4: The proposed model does not outperform other methods. What is the use-case for this model?__
> >
> > In fact, compared with task-specific models, we reduce the model development costs for different tasks. Compared to vanilla generalist models, we improve the training efficiency.
> >
> > Before generalist models, machine learning research usually develops task-specific models independently for different tasks.  Different task-specific models have their own specialized architectures, task formulations, training data, and training paradigms. These task-specific models also lack efficient collaboration among tasks. After pre-trained on task-relevant large-scale datasets, the model architecture and task formulation need to be re-designed and fine-tuned on sufficient downstream data. Additionally, each downstream task needs to replicate and maintain all parameters. These all bring expensive costs for new task development and limit the capability to meet the rapidly growing demands of diverse downstream scenarios.
> >
> > Take image caption as an example; typical task-specific models first pre-train the image encoders on ImageNet classification, and optionally pre-train the image encoders on Visual Gnome object detection. The text encoders are pre-trained with masked language modeling. After that, the image and text encoders are then jointly pre-trained on image-text datasets. Finally, they are fine-tuned on image caption tasks. Each training stage introduces task-specific architectures, task formulations, training data, and training paradigms. This complex training paradigm and potential design choices result in the expensive costs for new task development of task-specific models.
> >
> > The introduction of generalist models makes it feasible to process various tasks with a single model and shared parameters. Since all tasks share a set of parameters, applying generalist models to downstream tasks is less expensive. Generalist models can handle new tasks with unified task formulation. Specifically, single-stage multi-task pre-training combined with zero-shot inference and prompt-tuning paradigm can handle new tasks conveniently and efficiently, without the need to develop specialized models one by one.
> >
> > Nevertheless, all the existing generalist models follow the multi-task learning paradigm, where the task interference (or gradient conflict) caused by parameter sharing is inevitable and deteriorates task performance.  Our Conditional MoEs aim to solve this common problem in generalist models and further improve their training efficiency. By introducing Conditional MoEs, our models can achieve competitive performance with SOTAs using only <5% training data and <10% training cost, as shown in experimental results.
> >
> > ---
> >
> > > __Q5:  (a) The concerns about the limitation of attributes design. (b) How does token-level routing perform outside of the ablation study?__
> >
> > a) The attribute-level MoE does not contain any task-specific design and could generalize to unseen tasks by constructing attribute embeddings based on descriptions in Table 2. Its zero-shot performance is already verified in our experiments. Furthermore, new effective attributes can also be introduced into this framework to increase the expression of attribute description. We are also exploring more general ways to automatically generate attribute embeddings without pre-defining the attribute descriptions, such as the task2vec mentioned by Reviewer vm3E in Q1.
> >
> > (b) The token-level MoE may perform differently when the number of involved tasks increases or the datasets are much larger. However, in our current experiments, it does not show any advantage over the attribute-level version.  Besides, as stated in Table 3, token-level MoE has relatively higher training and inference cost, which is also a challenging problem when scaling to large-scale experiments.
> >
> > ---
> >
> > > __Q6:  The paper does not mention the Perceiver architecture.__
> >
> > We do not mention Perceiver because it is not a generalist model and there is not much connection between the Perceiver series (Perceiver and Perceiver IO) and Uni-Perceiver.
> >
> > The Perceiver series aim to build a general-purpose network architecture for neural networks that can handle data from arbitrary modalities. However, Perceiver still follows the task-specific paradigm, which needs to train different models for each task with different model sizes and parameters.  In comparison, Uni-Perceiver does not focus on the network architecture. The model architecture in Uni-Perceiver is just the vanilla Transformer encoder, not Perceiver.
> >
> > Nevertheless, thanks for the reminder; we will add the reference for Perceiver in our paper and mention its contribution in unifying the network architecture to process data from arbitrary modalities. We think this general network architecture could facilitate the development of future generalist models.
> >
> > ---
> >
> > >__Q7：Typo errors__
> >
> > Thanks. We corrected these typos in the submission.

---

### Official Review · Reviewer_vm3E · 2022-07-10

**Rating:** 7
**Confidence:** 4
**Soundness:** 3 good
**Presentation:** 3 good
**Contribution:** 3 good

**Summary:**

This paper proposes a generalist model (ie a model that handles multiple modalities and numerous tasks) based on a Mixture of Experts (MoE). The paper argues that, since current generalist models often share modules across different tasks and modalities, interference among different tasks is responsible for performance degradation when compared to specialist models. To mitigate interference, the proposed method dynamically routes the execution of specific tasks to specialist modules within the MoE.

**Questions:**

see Strengths And Weaknesses

**Limitations:**

see Strengths And Weaknesses

**Strengths And Weaknesses:**

- The proposed approach (a generalist Conditional MoEs) is well motivated. Different tasks can cause interference in gradient directions and lead to sub-optimal performance in individual tasks. Sparse mixture of experts, dynamically allocated based on a task prompt, has the potential to solve this problem while still retaining the ability to perform zero-shot task generalization.
- The existence of task interference was well-validated through an interesting analysis of image classification, masked language modeling, and captioning. It showed that while image classification and captioning can cooperate when learning lower layers in the model, they interfere with each other at higher layers.
- The paper proposes several approaches to conditioning the routing mechanism with task-related tokens (ie information that describes the current task and modality). The proposed routing strategies make sense, and they were shown to have an impact on the overall performance. It is was interesting to see that one of the best strategies was to rely on a hard-coded task description (summarized in Table 2). Would new tasks require new descriptions? Would the model benefit from a more generalized task description? Perhaps using learned task embedding, eg using something similar to task2vec ([https://arxiv.org/pdf/1902.03545.pdf](https://arxiv.org/pdf/1902.03545.pdf))?
- Experimental results (Table 3) show that interference has a detrimental impact on performance (as specialist models always have higher performance than a model which shares all parameters across tasks). The paper also shows that the proposed MoE is able to mitigate the impact of interference, achieving almost the same performance as specialist models on multiple tasks. Having said that, it is still disappointing that there is little evidence of task transfer, as the generalist model does not improve performance in most tasks (except for COCO Captioning B@4). I believe this is still a limitation that should be clearly stated in the paper.
- Finally, it is impressive that the proposed model was able to achieve competitive performance on several tasks (like object and action recognition, cross-modal retrieval and language understanding with little to no extra data. This ability to do zero-shot or few-shot transfer highlights the benefits of generalist models in general (which the current approach still retains).

In sum, I believe this is a strong paper. While the ideas of mixture-of-experts and the existence of interference in multi-task learning problems are not new, this paper successfully applies these concepts to generalist models, and achieves good results on several datasets.

---

> ### Author Response · Authors · 2022-08-02
> **Reply to Reviewer vm3E**
>
> We thank you for comprehensively summarizing the strengths of our work, and in particular providing useful suggestions to generate attribute embeddings in a more general way.
>
> ---
>
> > __Q1:  (a) Would new tasks require new descriptions? (b) Could the attribute descriptions be more general?__
>
> (a) The attribute descriptions in Table 2 used in our work do not have any task-specific design and can generalize to unseen tasks by constructing attribute embeddings easily. While we found that existing attributes are general enough to provide information for the tasks currently involved, adding new attributes should be beneficial when new attributes describe new tasks more comprehensively.
>
> (b) Thanks for your suggestion. We agree that the method used in task2vec is a more general way to generate task descriptions. We will explore how to automatically generate task attributes without introducing task ids in our future work.
>
> ---
>
> > __Q2: Why does it not improve performance in most tasks? Is there any task collaboration in the proposed model?__
>
> There is task collaboration in the joint training of generalist models. For clarification, we divide the tasks listed in Tables 3, 4, and 5 into 2 groups. (1) The first group includes tasks with large-scale datasets, such as ImageNet-21K and Books&Wiki. Such tasks have enough data, thus the knowledge from other tasks may not be helpful.  (2) The other group includes tasks with relatively small dataset, such as Kinetics-400 and MSCOCO caption. Those tasks have insufficient data and can benefit a lot from other tasks. While previous specialized models adopt the pre-training and fine-tuning paradigm to enable task collaboration, our generalist models only rely on joint learning to transfer knowledge across tasks. Even though the paradigm of pre-training and fine-tuning is well established to benefit from task collaboration, our generalist models can achieve competitive or even better results than those specialized models. This suggests that there is task collaboration in the joint training of generalist models.

---

### Official Review · Reviewer_LxEk · 2022-07-11

**Rating:** 6
**Confidence:** 5
**Soundness:** 3 good
**Presentation:** 3 good
**Contribution:** 2 fair

**Summary:**

The paper proposes several MoE routing strategies, i.e., token-level, context-level, task-level, modality-level and attribute-level on top of uni-perceiver, to mitigate task interference, showing noticeable improvements.


**Questions:**

1. Which conditional strategy is exactly adopted in the rest of the experiments? I see statement of `#param is the parameters required during model deployment' in table 4. I wondering if the adopted strategy is data-independent so that we can use only part of the whole parameters to do the inference.

2. What's the training FLOPS of both MoE and non-MoE version of the model?

3. How can such models do zero-shot inference on unseen tasks (in different form, instead of different domain) when task identifier is only id rather than general task representation like instruction? If using attribute condition, the 8-bit attribute seems also less expressive to be general.

4. How is the prompt tuning implemented? If the authors use exactly the same method with prior works, it's recommended to cite them or describe the detail in a simple way.

5. Just curious about how PT_{1%} compares to FT_{1%} and PT_{100%} to FT_{100%}.

6. OFA also reports the result of MS COCO caption using CE optimization only.




**Limitations:**

The authors have addressed their limitations and potential negative societal impact.

**Strengths And Weaknesses:**

\+ Different routing strategies, i.e., token-level, context-level, task-level, modality-level and attribute-level is evaluated. These strategies, though are quite common in their forms, are the first time being evaluated and proved effective in vision-language pretrained models, as far as I know.

\+ The paper has conducted extensive experiments, including without finetuning (WT), prompt tuning (PT), finetuning (FT), and zeroshot. All showing satisfactory results given its number of params and training data.

\+ The task interference problem is tested and shown as a clear motivation of using MoE.

\- The technical novelty is still somehow limited though it demonstrates effectiveness.

\- Some details of experiments are missing. e.g., It's not clear which conditional strategy is adopted at default, in all the experiments, it's not clear how the prompt tuning is implemented.

\- The term zero-shot in this paper seems to indicate data of unseen domain, since task id alone does not make it possible to achieve zero-shot inference for an unseen task in a totally different form.

---

> ### Author Response · Authors · 2022-08-02
> **Reply to Reviewer LxEk (1/2)**
>
> We thank you for the thorough review, noting the quality of our work.
>
> ---
>
> > __Q1: the default conditional strategy adopted after the ablation experiments.__
>
> As described in Line 258, we adopt the __attribute routing strategy__ after the ablation experiments, which can achieve promising performance and generalize to unseen tasks.
>
> ---
>
> > __Q2: What are the training FLOPS of the model's MoE and non-MoE versions?__
>
> The theoretical computational complexity (FLOPs) of training Uni-Perceiver-Base and its MoE variants on the ImageNet-1K dataset is listed below. We only report the FLOPS of forward pass for simplicity, and the empirical backward-forward FLOPs ratio for neural networks is roughly 2:1. Note that the reparameterization technique is utilized to merge top-K selected experts in the training phases for data-independent MoEs. Therefore, the training FLOPs for data-dependent MoEs will not increase much. However,  the reparameterization technique cannot be employed in data-dependent MoEs, so the FLOPs of these models increase significantly.
>
> |   | Un-Perceiver-Base | +token MoE | +context MoE | +modality MoE | +task MoE | +attribute MoE |
> |----------------|:-----------------:|:----------:|:------------:|:-------------:|:---------:|:--------------:|
> | Training FLOPs |       17.87G      |   29.02G   |    33.92G    |     18.04G    |   18.04G  |     18.04G     |
>
> By the way, we also compared the training time of different models in Table 3, which can also reflect the training cost difference of different models. We observed some inconsistency between training FLOPs and training time cost, which is probably because the communication cost is not considered in FLOPs calculation.
>
> ---
>
> > __Q3: (a) How can Uni-Perceiver-MoE conduct zero-shot inference on unseen tasks with different forms? (b) Is attribute less expressive to be general?__
>
> (a) Because attribute-level MoE does not have any task-specific design and parameters, it can easily construct attribute embeddings for new tasks by checking each description in Table 2, so that zero-shot inference can be conducted on unseen tasks with different forms by using the newly generated attribute. As shown in Table 5(c), our Uni-Perceiver MoE can achieve reasonable zero-shot performance on video-retrieval and video caption tasks, which did not appear in pre-training and have different forms compared to pre-training tasks
>
> (b) The attribute embedding is a flexible yet effective way to provide comprehensive information for multiple tasks. In our experiments, it can successfully maintain the generalization ability to unseen tasks. Furthermore, new effective attributes can also be introduced into this framework to increase the expression of attribute description. We are also exploring more general ways to automatically generate attribute embeddings without pre-defining the attribute descriptions, such as the task2vec mentioned by Reviewer vm3E in Q1.
>
> ---
>
> > __Q4: How is the prompt tuning implemented?__
>
> We illustrate our implementation of prompt tuning in Appendix L573-578. In short, following P-Tuning v2, we add learnable prompt tokens with random initialization at each transformer layer.  We also use linear heads with zero initialization for classification tasks.  Besides. the parameters in <SPE> token and the weight in LayerNorm layers are also tuned. The training receipts are listed in Appendix Table 9.
>
> ---
>
> > __Q5: the caption performance of OFA is lost.__
>
> Thanks for your reminder. __We have updated the caption result of OFA.__ In fact, the caption performance of OFA is referenced from https://arxiv.org/pdf/2202.03052v1.pdf, which does not include results using CE optimization only. The CE optimization results were added in https://arxiv.org/pdf/2202.03052v2.pdf on June 1st, which is after the submission deadline of May 19th. Therefore, only the caption result of OFA using cider optimization is listed in our submission.
>
> ---

---

> > ### Author Response · Authors · 2022-08-02
> > **Reply to Reviewer LxEk (2/2)**
> >
> > ---
> >
> > > __Q6:Comparison of prompt tuning and fine-tuning on 1% and 100% data.__
> >
> > The results in the original Uni-Perceiver paper have shown the superiority of prompt tuning under the 1% downstream data setting. Therefore, in our submission, prompt tuning is conducted when only 1% of downstream data is available, while fine-tuning is adopted on 100% of data.
> >
> > The results of these two tuning methods on 1% and 100% data for several typical tasks are listed below, from which we also found a similar phenomenon as in Uni-Perceiver: (1) Prompt tuning consistently shows better performance than fine-tuning on 1% data. We credit prompt tuning for tuning a few parameters and avoiding model over-fitting when data is insufficient. (2) However, prompt tuning may lead to under-fitting due to the limited capacity when there is enough data. Instead, fine-tuning adjusts all model parameters and requires sufficient data, which is a better option when 100% data is available.
> >
> > |  Task-dataset| Method|   PT(1%)  |   FT(1%)  |  PT(100%) |  FT(100%) |
> > |:-------------------------------:|:-----------------:|:---------:|:---------:|:---------:|:---------:|
> > |     Imagenet-1K (Accuracy )     |  Uni-Perceiver-B  |    80.9   |    80.4   |    81.7   |    84.0   |
> > |     Imagenet-1K (Accuracy )     | +Conditional MoEs |    82.0   |    81.3   |    82.5   |    84.5   |
> > |     Imagenet-1K (Accuracy )     |  Uni-Perceiver-L  |    84.2   |    83.1   |    84.8   |    86.2   |
> > |     Imagenet-1K (Accuracy )     | +Conditional MoEs |    84.9   |    83.4   |    85.2   |    86.4   |
> > |         K400 (Accuracy )        |  Uni-Perceiver-B  |    74.8   |    74.0   |    75.2   |    77.7   |
> > | K400 (Accuracy )                | +Conditional MoEs | 77.2      | 76.1      | 77.5      | 79.3      |
> > | K400 (Accuracy )                | Uni-Perceiver-L   | 80.0      | 79.2      | 80.3      | 81.9      |
> > | K400 (Accuracy )                | +Conditional MoEs | 83.0      | 82.4      | 83.2      | 84.2      |
> > | MS COCO Retrieval (i2t/t2i R@1) | Uni-Perceiver-B   | 68.4/51.9 | 67.5/51.9 | 68.9/52.6 | 69.8/53.9 |
> > | MS COCO Retrieval (i2t/t2i R@1) | +Conditional MoEs | 68.9/52.6 | 67.7/52.4 | 68.8/52.9 | 70.5/54.1 |
> > | MS COCO Retrieval (i2t/t2i R@1) | Uni-Perceiver-L   | 73.3/56.2 | 72.2/55.9 | 73.2/56.7 | 74.4/57.9 |
> > | MS COCO Retrieval (i2t/t2i R@1) | +Conditional MoEs | 73.3/57.1 | 72.6/56.5 | 73.0/57.5 | 74.7/58.3 |

---

### Official Review · Reviewer_RZvn · 2022-07-11

**Rating:** 6
**Confidence:** 3
**Soundness:** 3 good
**Presentation:** 3 good
**Contribution:** 3 good

**Summary:**

The paper presents an architectural improvement for attention based large scale task models. They start from the uni-perceiver, an attention based model to learn multimodal representations that is amenable to few-shot generalization via a prompting mechanism. The uni-perceiver has a token based modality conditioning and they 1) they introduce more flexible conditioning and 2) use that conditioning as inputs to a gating mechanism applied to the weights attention and MLPs in the uni-perceiver model. This is motivated by insightful but unsurprising experiments showing that in multi-task learning gradients coming from different tasks can have opposite directions thus specializing weights using a gating mechanism is meant to mitigate this effect. They show a broad set of experiments to demonstrate benefits of the new conditioning and the pretraining and generalization benefits of the approach.

**Questions:**

- the details on the gating could be improved. how do you train the gating ? is this more expensive in training than in inference ? Do you find collapse there or spurious gating solutions ? do you need to do something prevent these solutions ?

- how many experts are you using ? There must be a tradeoff between the number of experts and optimization ie with many experts less gradients are propagated through the other weights. do you have any insights that might help the readers ?

**Limitations:**

yes.

**Strengths And Weaknesses:**

strenghts:
- the CMoE idea and validation seems well executed
- the attribute routing seems reasonable and novel

weaknesses:
- some details are missing (see below).

---

> ### Author Response · Authors · 2022-08-02
> **Reply to Reviewer RZvn**
>
> We thank the reviewer for the suggestions and for recognizing the value of applying MoE to generalist models.
>
> ---
>
>
>  > __Q1：(a) What are the training details of the MoE gating? (b) How to avoid collapse for gating? (c) Is training MoE gating more expensive than inference?__
>
> (a) During training, the computation of MoE gating strictly follows Equation (3), all parameters are learnable. We mainly follow the training receipts for MoE gating in V-MoE[1] and Switch Transformer[2], which is illustrated in Appendix L568-572. In detail, following V-MoE[1], __a small normal noise is added to the gating activation of MoEs__, which has a standard deviation of 1/E. This noise encourages early gating exploration for experts and improves final performance.  We empirically found that the noise only affects the routing decisions for less than 5% of the training time. Besides, the balance loss in [2] is also employed for data-dependent MoEs to accomplish the balanced load of expert utilization.
>
> (b) __For data-dependent MoEs like token-level, it is important to use auxiliary losses to avoid the "collapse" phenomenon and encourage balanced loading for experts__. Specifically, the load balancing loss in Switch Transformer[2] is adopted in our models, which is multiplied by 0.01 and added to the total loss (see Appendix L570-572).  However, __for data-independent routing strategies like task-level and attribute-level, we found it is unnecessary to use balanced loss__, for which the collapse phenomenon has not been observed even without any auxiliary loss.
>
> (c) __For data-dependent MoEs (i.e., token-level and context-level), the forward cost is higher in inference than that in training__. The theoretical FLOPs of network forward in the training and inference phases are the same. However, the expert capacity factor is set to 1.0 for the training and 2.0 for the inference (see Appendix L568-572), which follows the common practice. The higher capacity factor in inference leads to lower computational intensity and higher communication cost, which is hardware unfriendly and results in higher forward cost in inference.
>
> __For data-independent MoEs (i.e., task-level, modality-level and attribute-level), the forward cost is higher in training than that in inference__. Since the gating decision of MoE layers in inference would be fixed for each task and does not need to be re-computed for each iteration as in training, the theoretical FLOPs of network forward is higher in training than that in inference. This results in higher forward cost in training.
>
> [1] Riquelme, Carlos, et al. "Scaling vision with sparse mixture of experts." Advances in Neural Information Processing Systems 34 (2021): 8583-8595.
>
> [2] Fedus, William, Barret Zoph, and Noam Shazeer. "Switch transformers: Scaling to trillion parameter models with simple and efficient sparsity." (2021).
>
> ---
>
> >  __Q2：(a) The default number of experts and (b) the influence when changing the number of experts.__
>
> (a) As illustrated in Appendix L568, we use __8 experts__ in all Conditional-MoE experiments by default.
>
> (b) In our experiments, increasing the number of experts will not introduce distinct optimization difficulty. We conduct experiments varying the number of experts.  As shown in the table below, the model will perform better when increasing the number of experts from 1 to 8. However, the performance gain reaches saturation when the number of experts exceeds 8.
>
> | Expert number   | Training time | Imagenet-1K | Imagenet-1K| COCO cap  | COCO cap |MLM|MLM|
> |-----------------|:-----------:|:-----------:|:-----------:|:-----------:|:-----------:|:-----------:|:-----------:|
> |                 |               | acc_train   | acc_val | acc_train | b@4  | acc_train | ppl_val |
> | 1 (dense model) | 1.00×         | 47.3        | 68.3    | 49.2      | 18.2 | 54.5      | 5.86    |
> | 2               | 1.12×         | 51.0        | 70.7    | 51.7      | 21.8 | 56.9      | 5.28    |
> | 4               | 1.26×         | 52.2        | 72.5    | 52.8      | 22.5 | 59.4      | 4.66    |
> | 8               | 1.43×         | 52.8        | 73.3    | 53.1      | 23.0 | 60.0      | 4.56    |
> | 12              | 1.57×         | 53.0        | 73.2    | 53.3      | 23.2 | 59.8      | 4.60    |
> | 16              | 1.79×         | 53.1        | 73.0    | 53.5      | 23.0 | 59.8      | 4.60    |
> | 32              | 2.42×         | 52.9        | 73.3    | 53.3      | 23.0 | 59.9      | 4.58    |
>
> We empirically found that, for attribute MoE, the gating decisions are usually stable and will not change rapidly after the early 5% training time. __Therefore, after early training, nearly all gradients will be propagated through fixed experts chosen for a specific task.__  Adding more experts will not increase the optimization difficulty when there are enough experts to separate conflicting tasks.

---

### Meta-Review · Area_Chair_rFp4 · 2022-08-26

**Recommendation:** Accept
**Confidence:** Certain

**Metareview:**

This paper proposes a generalist model Uni-Perceiver using conditional mixture of experts for processing multi-tasks. Most reviewers acknowledged the technical novelty of this work in CMOE, attribute routing, as well as extensive experiments. Reviewer KJqa argues the unclear affect of task interference for MTL. The authors reply well from a gradient perspective and experiments show task interference can be reduced. The other concerns about clarity in paper presentation and results are also addressed. The meta-reviewers thus recommend accepting it.

**Award:**

No

---

### Decision · Program_Chairs · 2022-09-14

Accept